# Lake Kinneret and Hula Valley Ecosystems under Climate Change and Anthropogenic Involvement

Moshe Gophen

Migal-Galilee Research Institute, P.O. Box 831, Kiryat Shmone 11016, Israel; gophen@migal.org.il

**Abstract:** The long-term record of ecological, limnological and climatological parameters that were documented in the Kinneret drainage basin was statistically evaluated. The dependent relations between environmental parameters and a change in climate conditions open a consequence dispute between three optional definitions: long-term instability, climate change impact and ecosystem resiliency. The Kinneret drainage basin during the Anthropocene era is marked by intensive anthropogenic involvement: Increase in population size, drainage of the wetlands and old lake Hula, agricultural development, enhancement of lake Kinneret utilization for water supply, hydrological management, fishery and recreation. Therefore, the impact of a combination of natural and anthropogenic environmental factors confounded each other, and the uniqueness of climate change is unclear.

**Keywords:** anthropogenic involvement; climate change; Kinneret; Hula Valley; nutrients; water level

## 1. Introduction

The temporal start of the Anthropocene Era, the youngest period within the global geological timetable was not officially aged. Nevertheless, eventually, it describes an Epoch when anthropogenic activity initiated a significant impact on climate and ecosystem structure and dynamics. Throughout the Anthropocene, the Hula Valley and Lake Kinneret are a compatible interlock of Anthropogenic achievement in the national economy, water supply, agriculture, nature protection and tourism management [1–3]. As of today, the integration between anthropogenic involvement and natural structure within the structure and dynamics of the ecosystem were mostly optimal. Nevertheless, future prediction requires precise knowledge about the role of man-made and climate change apart and "who is boss, and how much?" On one side of the ecological issue, there is a natural control of climatological and hydrological conditions, known as the commonly accepted concept of climate change. On the other side of the ecological issue, there is an Anthropogenic intervention. In Lake Kinneret: Dam construction enabling water level control, the construction of the National Water and Salty Spring Carriers, water and fishery utilization policy and others; In the Hula Valley: The swampy wetlands and old shallow lake drainage and agricultural development, Hula Project implementation and others. This paper is an attempt to evaluate priorities and optimization of the impact of those two eco-forces, climate change and Anthropogenic intervention aimed at future management design.

Lake Kinneret is the only natural warm mono-mictic freshwater lake in Israel which is located below sea level at the northern part of the Syrian-African, great rift valley. Since 1964, the lake has been the major source for domestic (drinking) water supply (350–450 mcm/year, $10^6$ m³ annually) in the country. From 2010, a major source of domestic water (650 mcm) has been from desalinization plants. Since the early 1950s, the Kinneret and the Hula Valley ecosystems have undergone significant modifications.

In 1933, a dam was constructed on the outlet of the lake, and lake water level became under human control. The fishery management and recreational activity became highly affected by lake water level. The lake ecosystem was a turnaround of nutrient availabilities from phosphorus to nitrogen limitation and frequent outbreak blooms of Cyanobacteria.

Anthropogenic involvement in the Hula Valley ecosystem initiated in the late 1950s. Until 1958, the Hula Valley was covered by the shallow Lake Hula (1.5 m mean depth; 1300 ha water surface) and 4500 ha of permanent and seasonally covered by dense vegetation swampy wetland cover [4]. During 1950–1957, the Hula Valley was drained, and the land use was converted into agricultural development. An implemented reclamation project (Hula Project) (1993–2007) included drainage canals renewal, newly created shallow Lake Agmon-Hula (110 and later 820 ha) and the establishment of portable spring-irrigation lines and renewal agricultural management. The drainage of the Hula swampy wetland and the old shallow lake was completed in 1957, and consequently, the very high variability of soil properties was indicated, and a study of the soil properties in relation to hydrological conditions became an essential demand. The function of the newly created ecosystem structure was the development of agricultural land utilization, ecotourism and water quality protection management. Consequently, agronomical and hydrological research is required to ensure the prevention of water quality deterioration in Lake Kinneret and efficient cultivation. Nevertheless, irrigation methods were inappropriate and were followed by damageable consequences such as underground fire and insufficient supply of irrigated water plants. During the second half of the 1990s, a reclamation project was completed, the Hula Project (HP), and agronomical utilization was significantly improved. One of the HP conclusions was to increase summer moisture of the cultivated land in the valley by higher Ground Water Table (GWT) and in the peat soil apart. The dynamics of nitrogen and phosphorus contribution by peat soil worthwhile additional moisture in summer and a supplement of irrigated water was allocated. Together with those anthropogenic achievements, the impact of fluctuated climate conditions became a significant factor in both the Hula Valley and Lake Kinneret management design. This paper is a tentative approach aimed at discriminating between anthropogenic and climate conditions' impact on the management design of the ecosystems.

*Study Area Site*

The most northern top of the drainage basin is 2814 masl located 61 km from Lake Kinneret (210 mbsl), creating a mean slope of 5%. Three headwaters, Hatzbani ($130 \times 10^6$ m$^3$/year), Banyas (app. $115 \times 10^6$ m$^3$/year) and Dan (app. $260 \times 10^6$ m$^3$/year), flow from north to south crossing the Hula Valley and join into one river, Jordan (app. $200$–$800 \times 10^6$ m$^3$/year). From the Hula Valley (61 masl), River Jordan flows along 15 km south into Lake Kinneret (210 mbsl). River Jordan supplies 63% of the Lake Kinneret water budget, of which other sources are direct precipitations, smaller rivers, runoffs and sub-lacustrine inflows.

Two geographical regions are focused on in this paper, Lake Kinneret (surface—169 km$^2$, volume $4 \times 10^9$ m$^3$, 210 mbsl, 26 m mean depth) and its drainage basin (2730 km$^2$), that of the Hula Valley (200 km$^2$ between 60 and 180 masl) (Figure 1).

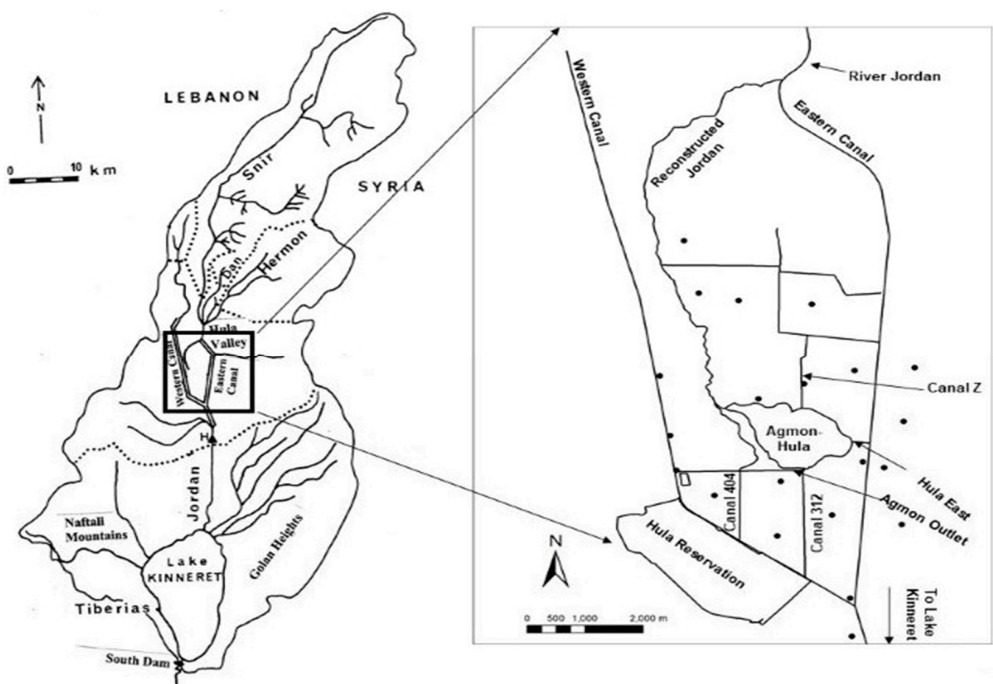

**Figure 1.** Geographical map of the Kinneret Drainage Basin (**left**) and the Hula Valley (**right**).

## 2. Material and Methods

Data about climate conditions were taken from annual reports of the following national services: Climatological conditions from the Israeli Meteorological Service; River discharges were taken from the National Hydrological Service, National Water Authority; Nutrient data in Lake Kinneret from the Kinneret Limnological Laboratory [5] and Mekorot Water Supply Company [6] Nutrient, in surface waters in the Hula Valley and hydrological data from the Hula Project Monitor program-Migal, Keren Kayemet LeIsrael and Water authority [7–10]. Air temperature data were supported by the Meteorological Service and Agriculture Service (Shoham; M. Peres). Statistical evaluation and data presentation was conducted using the software of STATA 17.0-Standard Edition, Statistics and Data Science, Copyright 1985–2021 StataCorp LLC, StataCorp, 4905 Lakeway Drive, 4905 Lakeway Drive, 800-STATA-PC, Stata license: Single-user perpetual, Serial number: 401706315938, Licensed to Moshe Gophen, Migal. Three statistical methods were utilized: Linear Prediction: A mathematical model where future values of a discrete time signal are estimated as a linear function of previous values. Lowess Smoother (LOWESS; Locally Weighted Scatterplot Smoothing) Regression Analysis creating a smooth line through a scatter plot to verify relationships between variables and trend anticipate; and Linear Regression: predicting the value of a variable based on the value of another variable.

*Chill Hours*

Temperature fluctuations measured at 15 stations located at altitudes ranging between 60 and 940 masl also confirm climate changes as shown in the record of accumulated "Chill Hours" carried out in the Kinneret drainage basin [2]: The long-term (1988–2021) record of annual Chill Hours accumulation is based on a modification of the "Chill Days Model" [11,12] and the "Utah Model" as follows: Air temperature (°C) is continuously monitored and hourly averaged; each hour with mean temperature below 7 °C is valued as 1; hourly temperature within the average range of 7.0–10.0 °C is valued as 0.5; mean range of 10.0–18.0 °C is valued as 0 and higher than 18.0 °C as −1; each 24 h are totally summarized into one number. If the total summary is a positive number, it indicates the additional Chill Hours for those 24 h. A daily record of Chill Hours obviously reflects air temperature changes. A daily Chill Hours report is practically carried out during the winter season (November through April).

The environmental factor of Albedo: This factor represents the following measures: If A = Total Incident Energy (sun and sky Radiation), and B = Energy Reflected from the Surface, then the Albedo definition is:

$$N = \text{Net surface Radiation} = A - B = \text{Albedo in \%.}$$

## 3. Results

*3.1. Regional Changes of Climate Conditions; Water Level Fluctuatiobns in Lake Kinneret, and Nutrient Migrations from the Hula Valley Twards Lake Kinneret*

For symptoms of climate change in the Lake Kinneret drainage basin, the following long-term fluctuated parameters were chosen: River Jordan discharge, air temperature, Lake Kinneret water level, nutrient migrations from the Hula Valley into Lake Kinneret, precipitation regime and number of rainy days and Chill Hours distribution. The results of temporal distribution of those parameters are given in Figures 2–15 and Table 1.

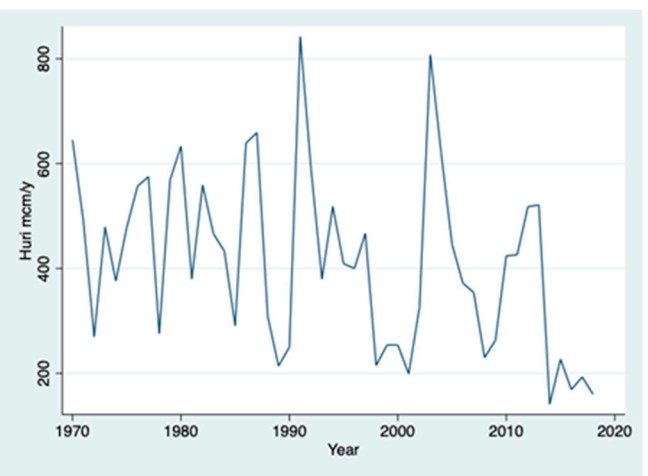
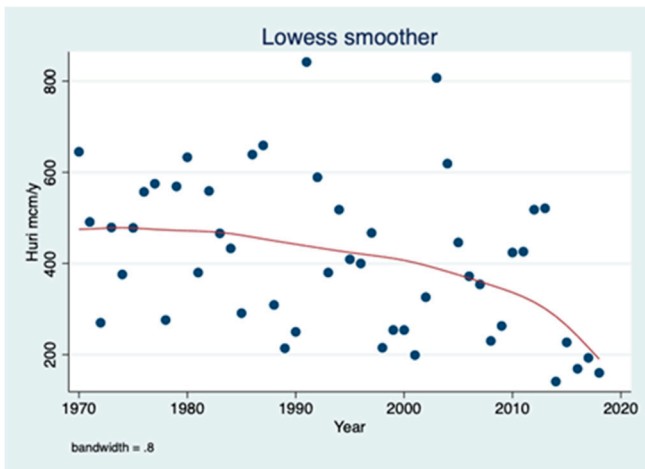

**Figure 2.** Signals of Climate Change in the Kinneret Drainage Basin are: moderate decline and distinct sharp decline in the discharge of River Jordan during 1970–2010 and 2010–2018, respectively; Jordan (Huri) annual discharge during 1970–2018 represents temporal fluctuated decline in precipitations: 7 "ebbs" representing drought (dryness) seasons and 8 "flows" accounting for flood seasons are shown.

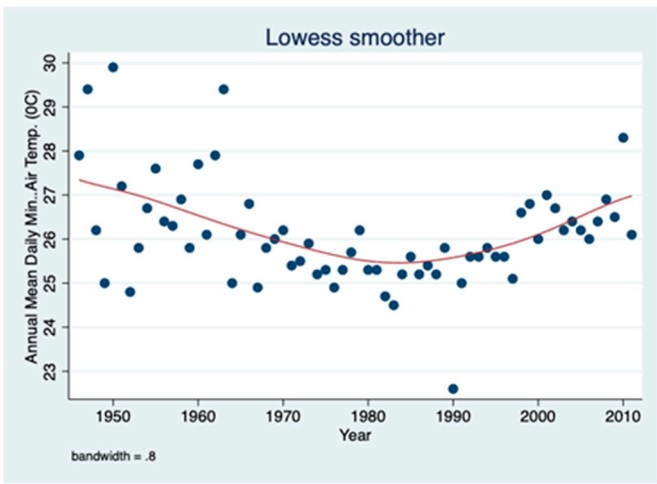
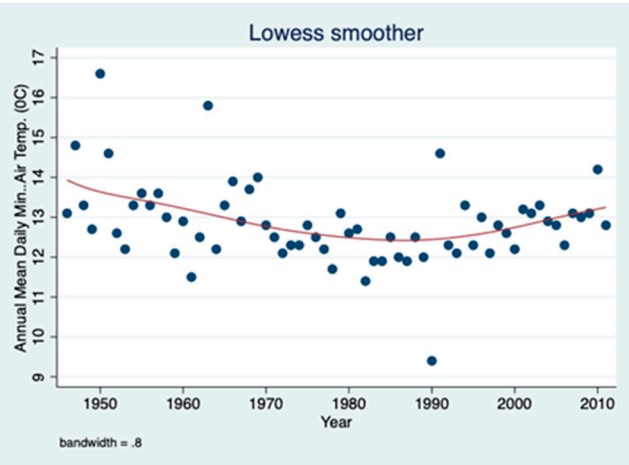

**Figure 3.** Annual averages of daily Maximum (**left**) and Minimum (**right**) air temp. (°C) (Dafna station, Northern Hula Valley) confirm distinct decline during 1940–Mid-1980s and onwards elevation of the maximum and minimum of daily temperature.

*Water Level Fluctuations in Lake Kinneret*

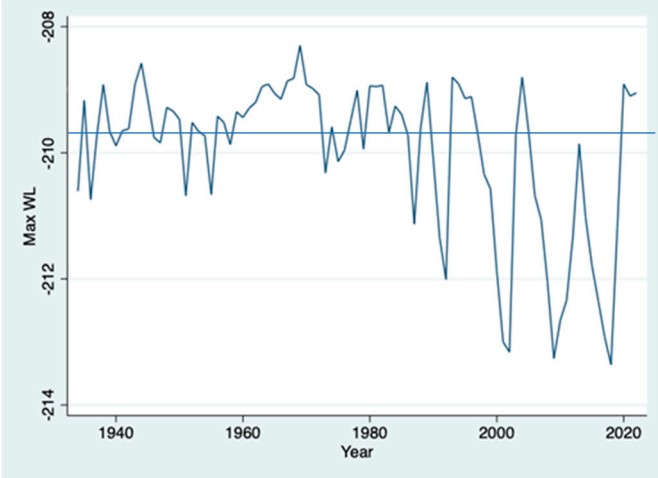

**Figure 4.** Line-Scatter plot of annual (1934–2023) highest WL (mbsl: meters below sea level) in Lake Kinneret. The total average is indicated (line: 209.99 mbsl; SD: 1.23 m). Values below and above the total mean indicate water scarcity and overwhelming, respectively: During 1933–1985, overwhelming, and later on, mostly scarcity.

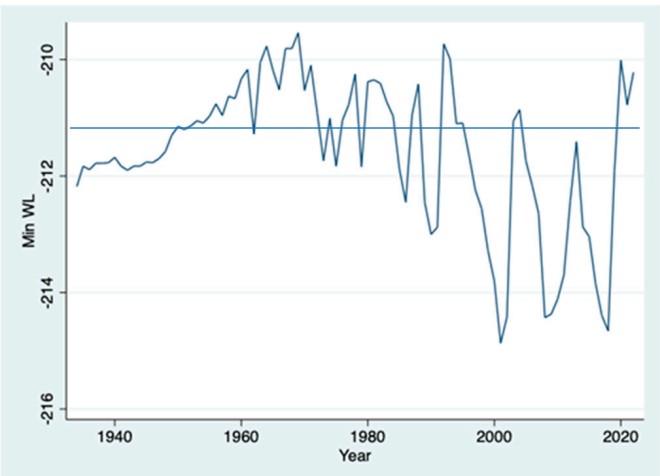

**Figure 5.** Line-Scatter plot of annual (1934–2023) lowest WL (mbsl: meters below sea level) in Lake Kinneret. The total average is indicated (line: 211.59 mbsl; SD: 1.28 m). Values below and above the total mean indicate water scarcity and overwhelming, respectively: Elevation of the lowest annual water level altitude during 1933–1970 and trend of decline afterwards are presented.

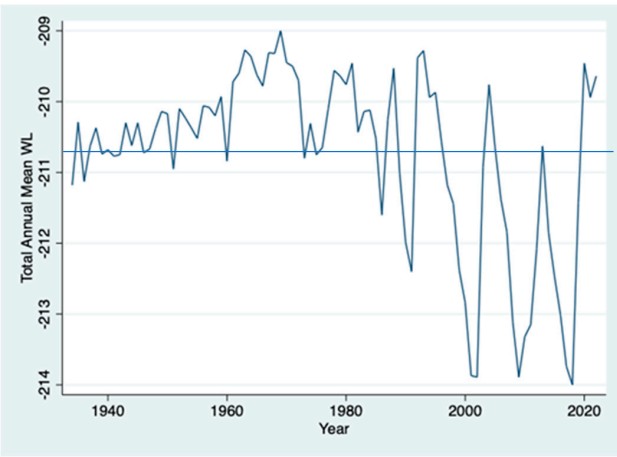

**Figure 6.** Line-Scatter plot of annual (1934–2023) mean WL (mbsl: meters below sea level) altitude: The total mean of 210.74 mbsl (SD 1.20 m) is lined: Continuous high WL in Lake Kinneret and a decline afterwards are presented.

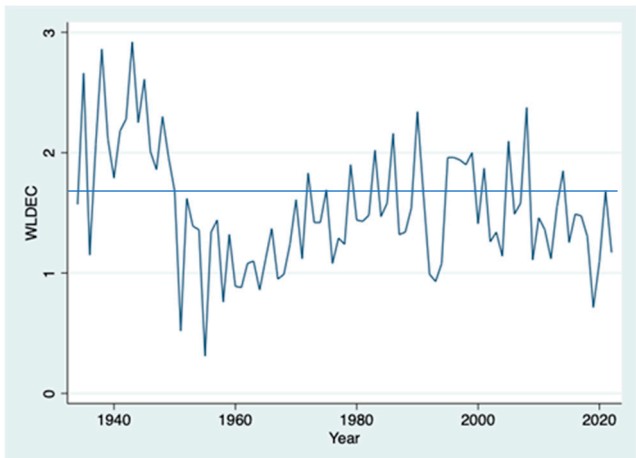

**Figure 7.** Line-Scatter plot of annual—actual (m) WL decrease in Lake Kinneret during 1934–2023. A total annual average of 1.54 m (SD: 0.51 m) (line): The annual decline was high above the total average, lowest during 1950–1970 and above the total mean afterwards.

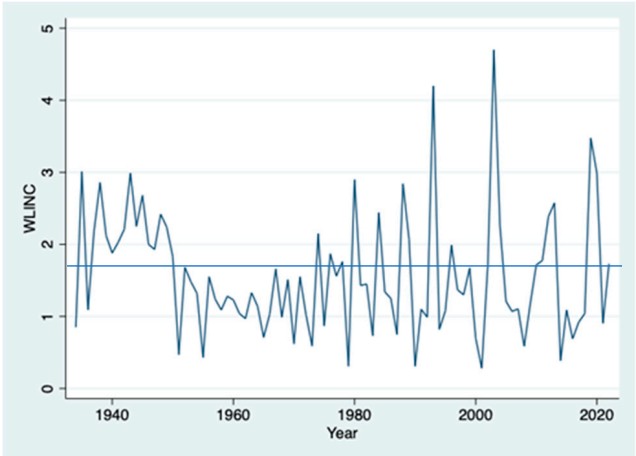

**Figure 8.** Line-Scatter plot of annual—actual (m) WL decrease in Lake Kinneret during 1934–2023. A total annual average of 1.58 m (SD: 0.84 m) (line): Periodical decline of above the total average during

1933–1950, and lower during 1950–1975 and mostly lower than the total mean afterwards (43 years) are presented.

*Nutrient Migrations from the Hula Valley towards Lake Kinneret*

The next Figures 9–15 represent temporal fluctuations in nutrient migration regimes from the drainage basin, through River Jordan towards Lake Kinneret.

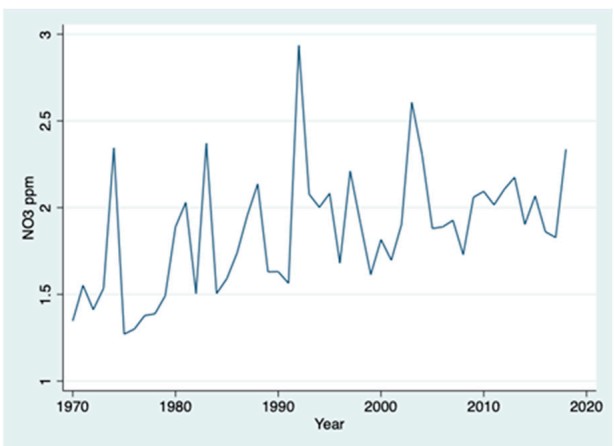
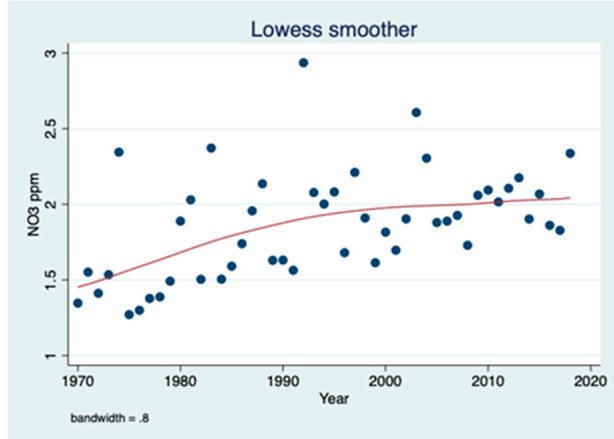

**Figure 9.** **Left Panel**—Line-scatter plot and **Right Panel**—Lowess Smoother plot of temporal (1970–2018) fluctuations in NO3 annual concentration (ppm; mg/L) in the River Jordan (Huri). Multiannual enhancement of NO3 concentration in the Jordan waters is presented.

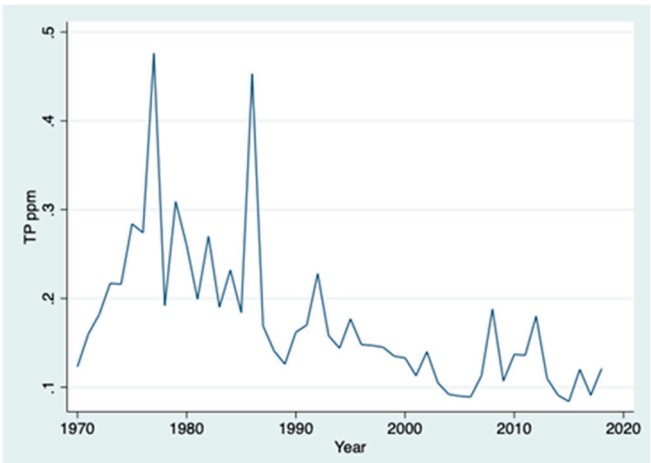
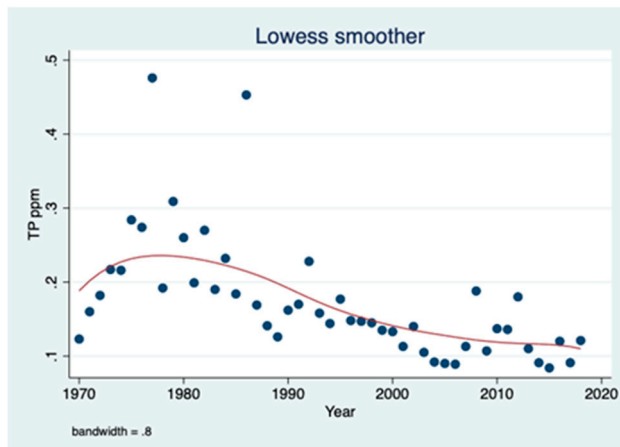

**Figure 10.** **Left Panel**—Line-scatter plot and **Right Panel**—Lowess Smoother plot of temporal (1970–2018) fluctuations in TP annual concentration (ppm; mg/L) in the Jordan River (Huri). Multiannual decline in TP concentration in the Jordan waters is presented.

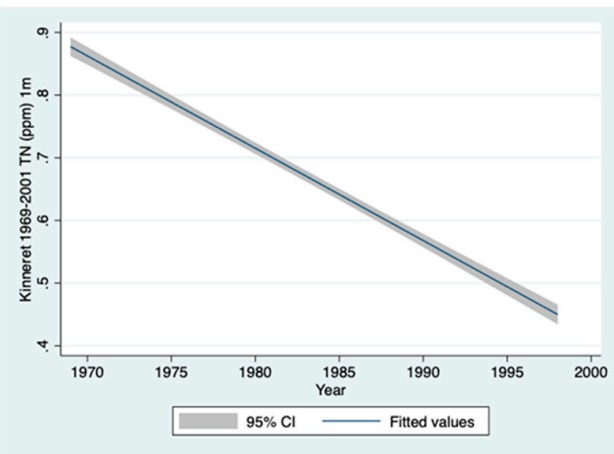

**Figure 11.** Linear prediction (w/CI; 95%) plot of temporal (1970–2001) changes in TN concentrations (ppm; mg/L) in the upper 1 m layer of Lake Kinneret. The temporal decline of TN concentration in the upper 1 m layer in Lake Kinneret is presented.

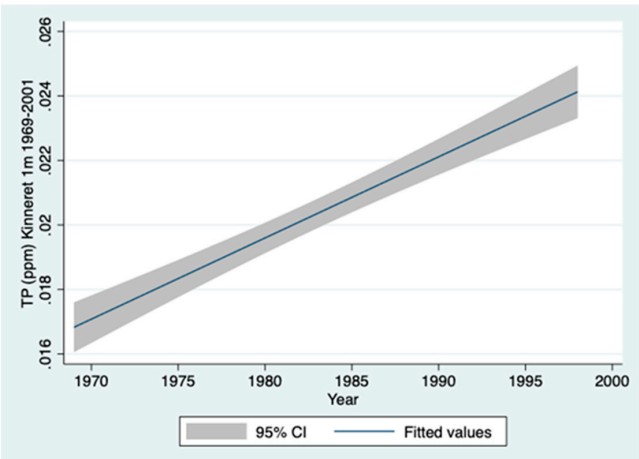

**Figure 12.** Linear prediction (w/CI; 95%) plot of temporal (1970–2001) changes in TN concentrations (ppm; mg/L) in the upper 1 m layer of Lake Kinneret. The temporal enhancement of TN concentration in the upper 1 m layer in Lake Kinneret is presented.

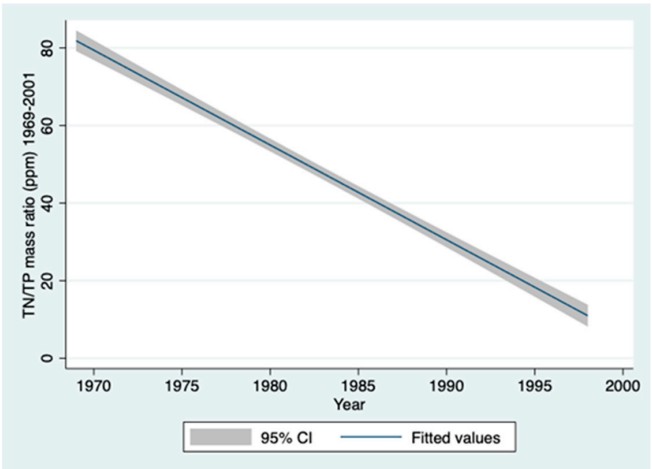

**Figure 13.** Linear prediction (w/CI; 95%) plot of temporal (1970–2001) changes in TN concentrations (ppm; mg/L) in the upper 1 m layer of Lake Kinneret. The temporal decline in TN/TP mass (concentrations ppm) ratio in the upper 1 m layer in Lake Kinneret is presented.

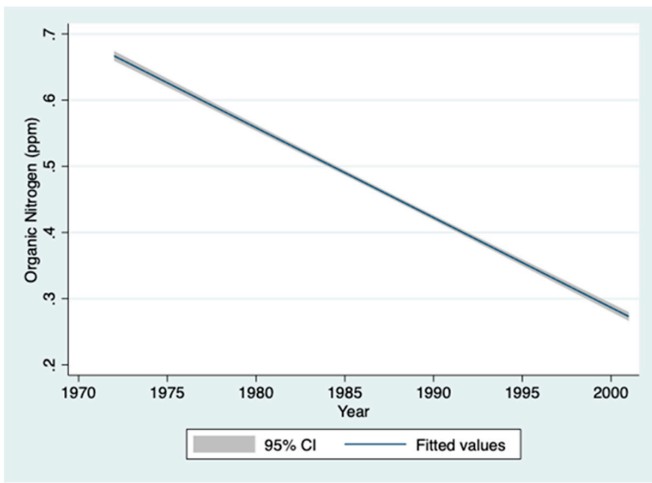

**Figure 14.** Linear prediction (w/CI; 95%) plot of temporal (1970–2001) changes in organic nitrogen concentrations (ppm; mg/L) in the upper 1 m layer of Lake Kinneret. The temporal decline in organic nitrogen in the upper 1 m layer in Lake Kinneret is presented.

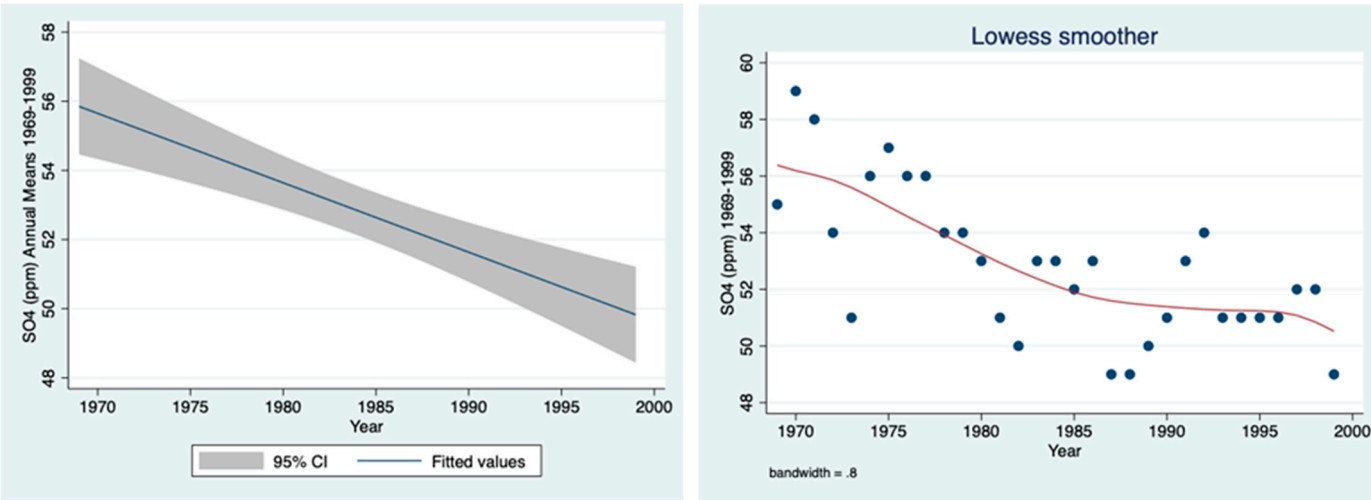

**Figure 15. Left Panel**—Linear Prediction plot (w/CI 95%) and **Right Panel**—Lowess Smoother plot of temporal (1969–1999)) fluctuations in SO$_4$ annual mean concentration (ppm; mg/L) in Lake Kinneret (all measurements, all depths, all stations and all dates). The long-term decline in SO$_4$ concentration in Lake Kinneret is presented.

### 3.2. Rain Capacities and Number of Rainy Days

Results given in Table 1 indicate a slight enhancement of precipitation capacity after 1968.

**Table 1.** Long-term record (1940–2021) of precipitation (mm/y) in Northern Hula Valley (Dafna station): Maximum (Max.), Minimum (Min.) and mean values (±SD; probabilities values (*p*) varied between 0.0009 and <0.0001) in three periods, 1940–1968, 1969–1991 and 1992–2021.

| Period | Max. | Min. | Mean (±SD) |
|---|---|---|---|
| 1940–1968 | 782 | 364 | 578 (±112) |
| 1969–1991 | 1038 | 358 | 632 (±177) |
| 1992–2021 | 1057 | 348 | 617 (±173) |

The annual mean of the number of rainy days during 1949–1980 was higher by 4.5% than the mean for the next period of 1981–2022. Rain capacity decline during the 2000s was accompanied by a minor decline in the number of rainy days.

The similarity between the temporal distribution pattern of an annual number of rainy days and capacity measure during the 2000s was indicated. It is therefore suggested that it is one of the climate change expressions.

## 4. Discussion

### 4.1. The Impact of the Albedo (See Methods) on Regional Air Temperature

The major step forward of the anthropogenic management of the Hula Valley was implemented in 1957 by the drainage of 6000 ha which were previously totally covered by water (old Lake Hula) and swampy wetlands occupied by dense vegetation. Thirty years later, 90% of this area was agriculturally developed except 10% water covered (fishponds and swamps residual).

Air temperature fluctuations in the Hula Valley are presented in Figure 3. The distinct impact of anthropogenic involvement, the Hula drainage, was reflected as mean air temperature declined by 1.3 °C (20.2 °C to 18.9 °C) from 1946 to 1990 [13]. It is suggested that before Hula drainage, when the valley was covered by water and dense vegetation, the air temperature was higher than during the post-drainage period resulted in the Albedo effect. The Hula Project (HP) implementation was initiated in the early 1990s and accomplished in 2006. During 1990–2010, air temperature was elevated by 0.9 °C [13,14]. Hula drainage was followed by a decline and HP implementation by an increase in air temperature. Albedo is the measure of the earth's surface radiation (heat) reflection. Water is much more absorbent and less reflective than soil, and bare soil is more reflective and less absorbent than grass-covered land. When the Albedo of a certain substrate is enhanced, heat balance becomes warmer and vice versa. Consequently, if Albedo increases, the air temperature above declines. It is suggested that when the Hula Valley was drained and the water cover area diminished from about 100% (before 1957) to 15.5% (after 1957) Albedo was elevated and the air temperature declined. During 1958–1986, the Hula Valley land was agriculturally cultivated with bare soil surface seasonality, which was later modified by the Hula Project implementation, and land grass cover was greener most of the time. This sequence of events caused the decline in Albedo accompanied by an air temperature increase. Therefore, air temperature fluctuation in the Hula Valley indicates the local–regional Albedo effect induced by the anthropogenic involvement (drainage, HP) effect and is probably not related to global climate changes.

Moreover, long-term (1988–2021) record of daily Chill Hours (see Methods) indicates a temporal (1990–2021) decline in the sum of cold hours, confirming the warming process.

### 4.2. Regional Hydrology and Kinneret Water Level (WL) Fluctuations [15]

The discharge of the River Jordan is fed by many water sources: Three major headwaters, smaller rivers, and surface waters, including the Golan Heights hydrological resources. Therefore, the regional approach within the study of the integrated impact of anthropogenic involvement and natural climate change is justified. The decline of the River Jordan discharge during the 2000s is confirmed (Figure 2). A question emerges: Who is dominant between two integrated mega-eco-forces of anthropogenic involvement and regional climate change in the Kinneret drainage basin? Moreover, due to the significant impact of anthropogenic involvement, regional climate change, was or was not? During 1934–1990, the annual mean Kinneret WL was above the multiannual average, whilst later onwards, in most of the years (31 years) the annual average WL was below the total mean. Consequently, the reason for that is anthropogenic involvement or climate change. Nevertheless, the management of the lake water balance, either storage or utilization policy, was not dictated with respect to the actual WL altitude. It was aimed at combining features of water quality protection and supplemental demands. How much was climate change dependent?

The Kinneret WL dynamics is a case where combined two mega-eco-forces of anthropogenic and climate conditions are integrated. Major constraints which directed the management of the Kinneret water level include national and regional factors significance: public awareness, fisheries, climate change and consequently water availability for supply, and water quality protection. Taking into account climate condition as the dominant impact on WL, evaluation of the long-term (1934–2023) record of the annual maximum WL altitude (Figure 6) indicates two distinct periods: (1) Fifty-one years (1934–1985) when maximal WL was mostly (7 exceptions) above the total average, and (2) Thirty-eight (1985–2023) years when maximal WL was mostly (8 exceptions) below the total average. It is suggested that regardless of anthropogenic factors, the climate change of rain and river discharge decline predominated this difference. During 1934–1964, the coherent continuation of minimal WL elevation represented management through dam, mostly or totally closed operations aimed at water storage policy. This storage policy enabled sufficient water supply for the electricity production southward at Naharaim station. In 1964, the National Water Carrier was inaugurated, and the hydroelectric production declined. Lake Kinneret became a national runoff water source for domestic supply. Consequently, the south dam was almost or totally closed and minimal WL was managed continuously high due to the water storage management policy. From the early 1980s and onwards, the impact on minimal WL management came through constraints of climate change and water supply demands. Exceptional cases occur during flood seasons (10 years). The annual mean WL history (1934–2023) (Figure 8) in Lake Kinneret indicates two distinct periods: 1934–1990 when mean WL was above the total average and 1991–2023 with WL measured below the total mean. It is probably a result of periodical change in climate conditions which confounded the impact of other factors. Evaluation of the annual decrease, increase and mean WL fluctuations in Lake Kinneret (Figures 4–8) during 1934–2023 reflects the factors (constrains) dominance priority affecting the annual fluctuations of decline and elevations of WL. During 1934–1950, the annual decline through open or half-open dams and no pumping was above the multi-annual average to supply hydroelectric energy for electricity production. Later on (1950–1990), the annual WL decline was enhanced gradually as the result of pumping intensification, whilst from the 1990s and onwards, the annual WL decline increased as a result of water source capacities' decline in the drainage basin, i.e., climate change, enhancing deficiency of available water. Data on annual WL increase (Figure 9) have shown that since 1950, when electricity production stopped, the annual WL increase was below the total average mostly affected by pumping and climate conditions, with 11 exceptional flood seasons.

Throughout almost the entire period after the mid-1990s, water was deficient (scarce), with few exceptional flood seasons, since dam construction (1933) and the National Water Carrier inauguration (1964) were affected by climate conditions (precipitation–river discharge regimes) and available water control altitude fluctuations in water level in Lake Kinneret.

During 1934–1948, anthropogenic control through dam regulation supported the operational demands for electrical production at the hydroelectric station southward but less than the yearly increase to ensure sufficient storage through continuous elevation of the minimal WL. After the operation of the National Water Carrier (1964) and the elimination of the hydroelectric station, the top priority of policy management became water storage, and accumulation for supply (pumping), resulting in low annual WL decline by outflow shortage (1950–1980) and, consequently, WL gradual increase. Nevertheless, as a result of continuous utilization, periodical WL decline was enhanced.

The Israeli Hydrological Service and National Water Authority confirmed the reduction in river discharges since the early 2000s. The long-term (1940–2018) record of precipitation (Dafna, northern Hula Valley) indicates a seasonal distribution pattern of wax and wane with ebbs and valleys (Table 1). Nevertheless, periodical trend differences are statistically insignificant. Rain capacity records represent high fluctuations amplitude which strongly affected the volume of available waters and, consequently, pumping supply regime, but long-

term trend changes are statistically insignificant. Moreover, supply requirements enhancement achieved the desalinization of seawater that was therefore implemented.

### 4.3. Climate Change and Nutrient Dynamics

The climate change, and independently the Hula drainage, enhanced peat soil moisture decline. The anthropogenic involvement through the Hula Project increased the peat soil moisture which was accompanied by enhancement of nitrate migration and increasing their concentration in the Jordan discharge.

Results presented in Figure 8 indicate co-occurrence of precipitation decline and consequently river discharge, and nitrate concentration enhancement. Elevated peat soil moisture in summer as part of the Hula Project recommended implementations of the Hula Project probably enhanced $NO_3$ migration. The enhancement of nitrate migration from the oxidized peat soil through the River Jordan discharge into Lake Kinneret as the result of anthropogenic involvement is therefore confirmed. The co-occurrence of precipitation decline and reduction in TP concentration decline is shown in Figure 4. Anthropogenic involvement through the Hula Project implementation enhanced summer wetting of the peat soil, but unlike nitrate, TP migration was diminished. The climate change, and independently the Hula drainage, enhanced peat soil moisture decline [16]. Nevertheless, the Hula Project implementation increased the peat soil moisture, which was accompanied by enhancement of nitrate but TP reduced migration.

The decline in organic nitrogen reflects the impact of both, anthropogenic involvement as Fish pond restriction and sewage removal, as well as climate change impact. The final result of external input load reduction and distinct decline in the lake ecosystem was clearly confirmed in the uppermost 1 m layer of Lake Kinneret (Figures 11–15). The impact of climate change integrated with anthropogenic intervention dictate water supply through water capacity availabilities, whilst water quality is dependent on nutrient dynamics. The hydrological factors within the climate change complexity are a key factor in an ecosystem, particularly in the Hula Valley. The Hula drainage enhanced soil nitrogen oxidation, and nitrate replaced ammonium as the major component within the Kinneret input loads. Ammonium migration from the unlimited stock that existed within the swampy wetlands was controlled by hydrological conditions [15]. Nitrates are loosely connected to the peat soil particles and efficiently disconnect and migrate through water flushing, rain in winter and irrigation in summer. Consequently, a hydrological trait of the peat-soil conditions defines nitrate migration from the Hula Valley into Lake Kinneret. The discharge capacity of the Kinneret inflow rivers is the factor that controls migrated load biomass and also their concentration. Significant linear regression existing between discharge capacity and $NO_3$ concentration was confirmed. Enhancement of the temporal (1970–2018) trend of $NO_3$ concentration in the Jordan waters resulted partly from the implementation of the Hula Project and Peat Convention, which recommended an increase in summer moisture of the peat soil. Unlike $NO_3$, TP migration from the peat soil in response to moisture elevation is the opposite: decline when moisture is enhanced and increase when wettability declines. The majority of the external migrated TP originates outside the Hula Valley, in other parts of the Kinneret drainage basin. Therefore, seasonal dryness during the 2000s (climate change) enhanced TP concentration in the Jordan discharge. Nevertheless, TP concentrations in Jordan waters are dependent on water-erosive eco-forces, and the decline in rain and river discharges reduced erosive energy impact. Conclusively, the climate change impact on TP migration is a controversial issue: enhanced geochemically vs. erosive decline in response to dryness conditions [16].

Nutrient concentration in the upper 1 m layer of Lake Kinneret is probably affected by external inputs supply. Intensive anthropogenic management in the Hula Valley during the 1980s was fishponds area drastic restriction and sewage removal, resulting in a decline in TN, organic nitrogen and TP input loads [17,18]. On the other hand, peat soil oxidation enhanced nitrate, and consequently TN, whilst the major source of TP supply from the drainage basin into the lake is located outside the Hula Valley. The summer migration

of TP from the Hula Valley is soil moisture-dependent, increasing moisture reduces TP migration. The integrated impact of climate change and anthropogenic involvement was therefore indicated: Wet climate enhances $NO_3$ and consequently TN, whilst dryness enhances TP migration from the peat soil in the Hula Valley. TN (mostly organic nitrogen; Figures 11 and 14) was declined as a result of fishpond restriction and sewage removal in the drainage basin, whilst TP (Figure 12) concentration at 1 m depth was enhanced as the result of external load increase induced by dryness effect. The overall impact was the TN/TP mass ratio (Figure 13) decline, which probably enhanced Cyanobacteria.

Major sources of $SO_4$ are sub-lacustrine salty-hot springs, gypsum dissolution in the Hula Valley and river discharge erosion. Consequently, climate change affected the external supply from the catchment and internal hot springs and sub-lacustrine vents. Anthropogenic involvement through irrigation and restructuring of the hydrological ecosystem (HP) in the Hula Valley also affected $SO_4$ inputs into Lake Kinneret. The decline in $SO_4$ (Figure 15) in Lake Kinneret therefore reflects both climate change and anthropogenic involvement. The major obstacle to achieving the climatological constraint of water scarcity and anthropogenic enhancement of domestic demands in Israel was an increased immovable factor of sea water desalinization production. Nevertheless, the occurrence of climatological extreme events and their impact on water availabilities are imposed by unpredictable conditions. It was documented recently in the Kinneret drainage basin: 2014–2018 dryness and 2019–2023 heavy rain regimes. Climate conditions are superimposed factors which are interlocked within the local anthropogenic constraints. These entail electricity production by hydroelectric technology, Lake Dam construction, national legislation of long-term utilization of Lake Kinneret as a drinking water source, and combined agricultural-eco-touristic land use in the Hula Valley. Conclusively, future modelling of climate change cannot replace an optimized management design which considers without priorities both natural climate condition and human properties.

The management of the Kinneret drainage basin, of which the Hula Valley is a significant part. This regional study area is a national concern with regard to water supply, population dispersion and their agricultural income resources efficiencies. Therefore, the comparative evaluation between anthropogenic involvement and natural climate condition changes is critical.

## 5. Conclusions

The Lake Kinneret and the Hula Valley ecosystems have undergone significant changes, resulting in ecological values. Climate changes, especially global warming, are widely documented. Nevertheless, regional changes in ecological traits are not distinctly documented within the Kinneret drainage basin. Evaluation of long-term records of ecological data within the Kinneret drainage basin defines equality in eco-forces initiated by anthropogenic involvement and climate changes. The unclear definition of the dominance value of each of these eco-forces requires data record continuation and a larger regional study area. The rationale of the paper is aimed at the uniqueness of anthropogenic involvement and climate condition, which require definite discrimination for appropriate management design.

**Funding:** This research received no external funding.

**Informed Consent Statement:** Not applicable as the study does not involve humans or animals.

**Data Availability Statement:** The data presented in this study are available upon request from the corresponding author.

**Acknowledgments:** Sincere appreciation is given to staff members for field data assistance and data evaluation of the Hula Project, Migal-Galilee Research Institute, Mekorot Water Co., Ltd., Lake Kinneret Data Base Kinneret Limnological Laboratory, IOLR Ltd. and The Kinneret and Drainage Authority for the Interim and annual reports availability.

**Conflicts of Interest:** The author declares no conflicts of interest.

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
