# Peer review of "Lake Kinneret and Hula Valley Ecosystems under Climate Change and Anthropogenic Involvement"

_climate, doi:10.3390/cli12050072_

Round 1

Reviewer 1 Report

Comments and Suggestions for Authors

The author has a kind of pretentious habit to use capital letters in words like "phosphorus", "water level". It is awkward, throughout all the manuscript. The grammar and style are poor. The manuscript looks like a technical report rather than a scientific paper. Some part of the text from Results should be moved to Material and Methods. In the results section the author show figures and comment all of them in a same way..."Results given in Figure X confirm/indicate etc. This style suggest that you gave proof in a scientific empirical or statitistical way. IF yes, please use such statements in Discussion section not in Results because it is a kind of interpretation. In my opinion the manuscript is lacking scientific style and my suggestion is to rewrite it. Some remarks in the attached pdf.

Comments on the Quality of English Language

The author has a kind of pretentious habit to use capital letters in words like "phosphorus", "water level". It is awkward, throughout all the manuscript. The grammar and style are poor. The manuscript looks like a technical report rather than a scientific paper.

Author Response

Response to Reviewer #1:
1) All figure captions were revised. The revised version is suitable to the chapter of Results

Reviewer 2 Report

Comments and Suggestions for Authors

Comments

The article certainly deserves the attention of specialists and can be published in the journal. The material is interesting as a description of the problem of local climate change.

1. Paragraph 3.Results. Descriptions of the figures (Fig. 3 and below) are given after the drawing itself. Descriptions and comments should be placed before the drawing.

2. Figures 2, 3. The headings say “Climate change”. These graphs do not show climate changes, but only changes in parameters: annual discharge during 1970-2018 or Annual averages of …. air temp. Only what is shown should be signed.

 3.      Figures 4–8. What do the numerical values under the figures “–209.99(±1.23)” and others mean? If this is a “total average”, then they must be noted in the figure caption, and the line with these values must be marked on the graphs.

4.      Figures 4–6. In what units is the level measured? In cm or in m? Only in Figs 7 and 8 it is noted that in meters. In hydrology, it is common to use centimeters. Incorrect designation misleads the reader. The situation is the same in other parameters. It is necessary to clarify the units of measurement of the scale.

 5.      Figures 9, 10. The headings read “The impact of Climate Change”. But these graphs do not show the impact of climate change on anything, but its show changes in parameters: “Annual mean concentrations of NO3” or “annual mean of TP concentration”. Only what is shown should be signed.

 6.      Figure 9 shows an increase in concentrations of NO3, and Figure 10 shows a decrease in TP concentration. But Figures 11 and 12 show the opposite trend. Why?

 7.      The description of the methodology and the Results do not contain a description of the Linear prediction methods. How were the calculations made, what model or software was used?

 8.  It may be worth combining the Results and Discussion sections. And give a more detailed step-by-step description of the results obtained.

 9.  The author connects changes in the level and removal of nutrients with temperature changes and anthropogenic impact. But the figures do not show when and what phenomenon occurred. It is possible that a time scale of anthropogenic events should be added to the graphs. It is also necessary to combine temperature graphs with level fluctuations and the removal of nutrients. In addition, add data on the amount of precipitation, since this should also determine the volume of water.

Author Response

Response to Reviewer #2:
1) The location of figure below (not above) its figured illustration is commonly accepted in scientific documentary as in this paper.

2) Temporal changes of temperature and precipitation are obviously climate conditions . Therefore the subtitle was revised to: Changes of climate condition.

3) The total averages of mbsl were precisely inserted properly into the caption of Figures  3-5 and a line was indicated in the illustration respectively.

4) The captions of figures 6,and 7 was revised and the value in m units was included.

5) The vertical axes units of nutrient concentrations in Figures 8-14 are all presented properly as ppm (parts per million) equal to mg/L which are both used in scientific documentary.

6) Climate change version in Figure 9 was revised to climate condition.

7) Figure 9 illustrating Jordan water concentration and figures 10-14- those of Lake Kinneret therefore – no contradiction.

8) Description of statistical method was added to the Method chapter.

Reviewer 3 Report

Comments and Suggestions for Authors

General comments

The introduction should report more details with respect to the study objective, but also a broader framing of the problem although it is a fairly unique case study.

It is suggested that the description of the study area be treated separately from the introduction, starting with "Two geographical regions are focused ..." adding a paragraph "Study area"

The author has long experience in studies of the area under consideration, but for this he should facilitate the reading with a more careful and detailed description that will enable the reader to better understand the pressure factors present. In addition, a central aspect is the Hula Project but its description is too ommary and spread over several parts of the manuscript. This does not contribute to understanding the overall results and the realtion between climate change and observed phenomena.

Detailed notes

Unit of measurement

It is surprising that we still need to point out the use of units reported in a manner inconsistent with the International System (SI) of units.

E.g.:

- mcm is NOT an accepted unit but is spelled m3

- Between106 and m3 there always goes a space

- if you a the SI you do not use other units outside the system (ha)

- masl is spelled m asl and so is mbsl is spelled m bsl

- also ppm has not been used in water chemistry for years, but mg/L is used

- in several graphs on the ordinates the unit is missing

these are some of the errors that need to be corrected, but a careful reviation of formalisms is urged.

Graphs and statistics

Regressions are shown in various cases without statistical indicators (e.g., Figs. 2, 9, 10 and 14(which is actually 15, because there are 2 Fig. 14!) on the right.The same is true for Figs. 11, 12, 13 and 15 (shown as 14). Talking about the meaningfulness of these regressions helps to understand the goodness of the interpretive model used.

Graphically it would be better to include in the graph box the mean values placed below Figs. 4, 5, 6, 7, 8, but this is just a matter of aesthetics.

Rather, one does not understand the use of "scatter plots" for a representation of lines connecting points. 

Finally, with the SDs shown, the differences in the averages in Tab 1 are actually not statistically significant. More elaboration would be needed on the significance and at what level (0.0

Various errors

In the manuscript there are minor editing errors, Eg )of without a space, migrationbs fromn, gven, ALBEDO and Albedo (use albedo), removal. and, annual eincrease, ...

However, it should be noted that there is an improper use of the IUPAC formalism: the names of elements and compounds are in lower case!

Conceptual aspects

The Author sometimes uses unclear terminology. For example the use of "nurient migrations from the Hula Valley" could be changed to "nutriens loads" more used in nutrient balances in lakes. Again: what does "toal mean" mean? Or "minimal" and "maximal" for minumum d maximum, or "Rain capacity" whose meaning is not understood.

Then there is the case of albedo, whose conceptual definition is not well explained. In fact, the albedo is an index that varies from 0 to 1 and, therefore, is the ratio between the net surface radiance and the total incident energy. Furthermore, it is a dimensionless index and not a %, or at least it is usually indicated as such. Finally, we talk about "precipitation decline" and cite fig. 8, which represents the variations in the lake level. It may be that the Author meant that the decrease in precipitation lowers the lake level, but it is not at all clear in the text.

Final remarks

The manuscript needs a thorough revision because it is currently unpublishable.

The review suggests:

- better describe the general context,

- eliminate the various contradictions that lead to a lack of understanding of the final conclusions and, finally, frame the results in a more general context.

PS The overemphasis on the Anthropocene is misleading.

Comments on the Quality of English Language

The quality of English Language require e revision for several mistakes in the manuscript.

Author Response

Response to Reviewer #3:

  • Accepted and implemented
  • Accepted and implemented
  • Information about background constrains and objectives were added.
  • All units in this paper (climatological parameters, chemical concentrations, river discharge. actual and altitude WL change in m, and masl, mbsl respectively) are in accordance with the international SI system units/
  • Ful meaning of ppm concentration is given in Methods chapter: ppm=parts per million =mg/L as given in scientific documentary.
  • Full lettering and meaning of mbsl and masl are given in Methods chapter.
  • Mg/L equal ppm (parts per million. As totally accepted in scientific documentary.
  • Figures numbering was revised and corrected, Statistical indicators are built-in and presented within the illustration (w/CI 95%, grey color).
  • Scatter plot was changed to Line-Scatter plot as line connection between dots (observations). Table 1: probability (p) of all tested parameters were significant (S) with p values range between 0.0009 - <0.0001, commented in the text.
  • ALBEDO was changed to Albedo throughout the entire MS.
  • Migration is different from load: migration is exit from soil sources; load=input.
  • Changes were inserted: Total mean=mean of the totals; Maximal=maximum; minimal=minimum; Rain capacity=total annual precipitation in mm.
  • Obviously: Rain decline was followed by WL decrease.

Round 2

Reviewer 3 Report

Comments and Suggestions for Authors

The manuscript has been partly revised as suggested.

However, some deficiencies remain:

- the presentation of the results is only graphic and there is a lack of reflection on their representativeness of the studied area

- the observation on the definition of albedo that if % is given then it cannot be expressed by the dola difference between A and B has not been taken into account. Also, I would move this methodological discussion and its important significance into methods,

- the sentence should be better explained in the conclusions: Evaluation of long-term records of ecological data within the Kinneret drainage basin defines equality in eco-forces initiated by anthropogenic involvement and climate changes. Indeed, it is not clear why and what data collected lead to this consideration. Rationale I would put in the discussion.

Finally:

- the title of section 3.5 is unnecessary, because the text describes the data in the table above

- there are two captions of Fig. 14!

- it is grammatically incorrect to use Nitrogen and Phosphorus, because they should be spelled (IUPAC) nitrogen and phosphorus! 

Comments on the Quality of English Language

No particular comments: please submit the text to have a control by a mother language

Author Response

Author response to Reviewer #3 3rd Round

  1. Comment accepted: explanations about the relevance of the presented data to the study area and its environmental trait was extended respectively.
  2. The difference between A and B in the Albedo phrases express as percentage. It is commonly used in relevant scientific documents.
  3. Comment accepted: A text was added to the discussion to extend the background relevance of the paper focused rational point of the impossibility to discriminate properly between anthropogenic and natural (climate condition changes) impacts, both are significant.
  4. As response #3 a sentence was added to the summary with regard to emphasize the focus of the paper.: comparative consideration between anthropogenic involvement and climate change.
  5. The title of section 3.5 was relocated.
  6. One of the Figure 14 caption was eliminated.
